# Successful Resection of Retrobulbar Carcinosarcoma without Recurrence: A Case Report

**DOI:** 10.3390/medicina58020317

**Published:** 2022-02-19

**Authors:** Chun-Hao Huang, Lung-Chi Lee, Hong-Wei Gao, Yi-Hao Chen, Ke-Hung Chien

**Affiliations:** 1Department of Ophthalmology, Taichung Armed Forces General Hospital, Taichung City 41168, Taiwan; fun_creater@hotmail.com; 2Department of Ophthalmology, Tri-Service General Hospital, National Defense Medical Center, Taipei 114, Taiwan; kidday0205@gmail.com (L.-C.L.); doc30879@mail.ndmctsgh.edu.tw (Y.-H.C.); 3Department of Pathology, Tri-Service General Hospital, National Defense Medical Center, Taipei 114, Taiwan; doc31796@gmail.com; 4Department of Pathology, Tungs’ Taichung Metroharbor Hospital No. 699, Section 8, Taiwan Boulevard, Wuqi District, Taichung City 43503, Taiwan

**Keywords:** orbital carcinosarcoma, tumor resection, radiotherapy, recurrence

## Abstract

Carcinosarcomas are biphasic tumors comprising carcinoma and sarcoma components that occur in many tissues but are rarely found in the orbit. A 70-year-old male presented to the ophthalmic clinic with progressive proptosis, having decreased vision in the left eye for 8 months. On examination, severe exophthalmos and lagophthalmos with limited extraocular movement were noted. Orbital computed tomography scans revealed a large, well-defined, heterogeneously enhanced mass in the left retrobulbar orbital cavity. The tumor was completely resected, and the pathological examination revealed a carcinosarcoma. The prognosis was excellent without local recurrence at 48 months postoperatively. Thus, when considering treatment for effective management of such tumors, tumor resection followed by radiotherapy or chemotherapy is highly recommended.

## 1. Introduction

Carcinosarcoma, also referred to as a true malignant mixed tumor, is a biphasic tumor with carcinoma and sarcoma components. A monoclonal origin is considered with myoepithelial cells to be the most likely candidate for common stem cells [1]. Carcinosarcomas occur in many tissues, including the uterus [2,3], bladder [4], skin [5], gastrointestinal tract [6,7], lungs [8], and organs of the head, including the salivary gland [1], lacrimal gland [9], and paranasal sinuses [10]. There are five previously reported cases of primary orbital carcinosarcoma [9,10,11,12], and only one case showed no recurrence for 10 months postoperatively. Although it is aggressive and exhibits both local tissue invasion and metastasis, we report the successful treatment of an orbital primary carcinosarcoma using total excision and radiotherapy without recurrence 48 months following treatment.

## 2. Case Report

A 70-year-old Asian male presented to our department with progressive proptosis. He complained of decreased vision in the left eye for 8 months. The patient had no known systemic disease and had undergone cataract surgery in his left eye 3 years previously. 

On examination, the presenting uncorrected visual acuity (VA) was 6/6 in the right eye and counting fingers at 30 cm in the left eye. Intraocular pressure (IOP) was 13 mmHg and 31 mmHg in the right and left eye, respectively. The relative afferent pupillary defect was negative in both eyes. 

Evidence of exophthalmos, hypotropia, and lagophthalmos of the left eye (Figure 1) was noted with Hertel values of 20 mm and 27 mm in the right and left eye, respectively. Extraocular movement was limited in the left eye, especially in the medial gaze position; slit-lamp examination of this eye revealed exposure keratopathy induced by lagophthalmos. 

The computed tomography scan of the orbit revealed a heterogeneously enhancing mass in the left retrobulbar orbital cavity approximately 3.7 × 2.5 × 2.5 cm in size involving the frontal bone and extending to the left frontal sinus, causing compression of the left superior rectus, lateral rectus, and inferior rectus muscles, as well as anterior and downward displacement of the left eyeball resulting in exophthalmos (Figure 2).

Lateral orbitotomy was performed to access the retrobulbar space. Frozen sections revealed malignancy accompanied by infiltrating anaplastic spindle tumor cells that had hyperchromatic, pleomorphic nuclei exhibiting frequent mitoses at a distance of 1 mm from the nearest surgical margin. A heterogeneous tumor, measuring approximately 4.0 × 3.0 × 3.0 cm, was carefully dissected, and total resection was performed from the retrobulbar space.

Pathological examination revealed a carcinosarcoma (Figure 3A) characterized by mixed adenocarcinoma and sarcomatous components (Figure 3B) showing pleomorphic and hyperchromatic tumor cells, atypical mitoses, and focal tumor necrosis. The glandular tumor cells and focal sarcomatous cells were positive for pan-cytokeratin staining (Figure 3C). Immunohistochemical staining for vimentin demonstrated a negative result for the glandular tumor cells and positive immunoactivity for the sarcomatous component (Figure 3D).

The patient was subsequently referred to a radio-oncologist for systemic survey and radiation therapy. Although positron emission tomography images revealed no metabolic evidence of systemic malignancy 1 month following the operation, the patient received adjuvant radiation therapy owing to the high local recurrence and distant metastasis rates.

Two months following surgical resection, the patient underwent intensity-modulated radiation therapy over a course of 21 sessions during which the tumor bed and left orbital cavity were subjected to a total dose of 40 Gy (195 cGy per fraction) using 6 MV photons, 5 times a week. The patient was originally scheduled to receive a total dose of 50 Gy over 26 sessions of radiation therapy; however, the treatment was terminated early owing to intolerance to its side effects including dry eye and conjunctivitis. Orbital magnetic resonance imaging and bone scan was performed postoperatively at 3 months and 12 months and yearly thereafter, which revealed no local recurrence of the tumor at 48 months postoperatively. At the last follow-up, 48 months following the tumor resection, the VA remained counting fingers at 30 cm, and the IOP was 13 mmHg in the left eye.

## 3. Discussion

Carcinosarcoma, a true malignant mixed tumor, is an intermixed biphasic tumor with carcinomatous (malignant epithelial) and sarcomatous (mesenchymal) components. Two theoretical hypotheses have been proposed to explain the histogenesis of these tumors. The first is the multiclonal hypothesis (convergence hypothesis), which supports that the tumors originate from two or more stem cells, the epithelial and mesenchymal components, and should be considered to be a type of collision tumor. The second is the monoclonal hypothesis (divergence hypothesis), which proposes that the tumors stem from a single totipotent stem cell that differentiates into separate epithelial and mesenchymal components. The findings of a study by Thompson et al. showed that six carcinosarcomas from different organs, including the uterus, breast, lung, and gastrointestinal tract demonstrated monoclonal origin from common totipotent progenitor cells [13], which concurred with the monoclonal hypothesis. 

Nevertheless, the convergence theory (sarcoma derived from metaplastic transformation of the carcinoma) has also been favored over the combination theory (both components stemmed from the single stem cell clone) in carcinosarcomas of the female genital tract [14], including the uterus [15] and ovary [16], as well as that of the salivary gland [17]. The transformation of carcinoma to sarcoma in these tumors resulted in sarcomatoid carcinomas, which were then termed carcinosarcomas when they demonstrated well-defined differentiation to sarcomatous tissues. The sarcoma and carcinoma components of these tumors both shared the same genetic alteration and were monoclonal, which has been demonstrated in patterns of clonality, genomic analysis, and loss of heterozygosity [18].

There are three types of malignant mixed tumors, the most common type of which is carcinoma ex pleomorphic adenoma, which arises from a pre-existing pleomorphic adenoma. The second type is metastasizing mixed tumor that has benign-appearing epithelial and stromal components, and the third is carcinosarcoma, a true malignant mixed tumor [16] that is extremely rare and has both malignant epithelial and stromal elements. In our case, the tumor could have originated from a benign mixed tumor (pleomorphic adenoma) and subsequently transformed into a carcinosarcoma [19], which is rare. Pleomorphic adenomas that are commonly observed in the salivary gland are benign tumors that carry a risk of malignant transformation of approximately 1.5% in 3 years and 9.5% in 15 years since the time of tumor occurrence [17]. While many authors have assumed that myoepithelial cells were common progenitor cells of tumors, including those of the breast [20] and salivary glands [21], others have been unable to identify this type of cell as a common progenitor [22]. Risk factors for the transformation of urothelial carcinoma into carcinosarcoma, including previous radiation therapy, chemotherapy (e.g., cyclophosphamide), or other conditions that cause cell replication abnormalities, have been reported [23].

Carcinosarcoma of the orbits can be of either primary or metastatic origin. Metastases of carcinosarcomas to the parotid gland [24], breast, prostate, lung [25], and uterus [26] have previously been reported. A study reviewing carcinosarcomas of the maxillary and ethmoid sinuses, with extension into the orbits, has been conducted [10]. Furthermore, a case of recurrent cutaneous carcinosarcoma of a previously resected basal cell carcinoma originating from the medial canthal region has also been reported [27]. 

Five case reports describing primary carcinosarcomas arising in the orbit have been reported [9,10,11,12], of which only three case reports provided sufficient evidence for the existence of a carcinosarcoma. Among the five cases, one patient showed no evidence of recurrence 10 months following surgical resection and heavy particle beam therapy [9], one patient refused surgery and died 13 months after symptom onset [10], another patient developed lung metastases [12], one patient was lost to follow-up following surgical resection [10], and no follow-up information was available for the final case [11]. Our case has the longest documented survival time of 48 months, with no evidence of recurrence following total surgical resection and subsequent radiotherapy.

Further, the tumor in one previous case arose from a pleomorphic adenoma of the lacrimal gland [9] and those in two patients arose in the accessory lacrimal glands and submucosal glands in the area of the lacrimal sac [10]. In another two patients, the tumors originated from the lacrimal glands [11,12], although sufficient details regarding these were not available. Although the tumor in our case appeared to arise from a pleomorphic adenoma of the lacrimal gland, based on the histopathological features, it could have originated in the left frontal sinus. Among the patients of these previous cases, four were female and the sex of one was not reported [9,10,11,12]; ours was the only case pertaining to a male patient. 

Owing to the rarity of orbital carcinosarcomas, there is no consensus on the management and prognosis of such tumors, and prospective trials have not yet been performed. Based on the reviews on uterine carcinosarcomas, the recommended mode of management is complete surgical resection with subsequent systemic chemotherapy, including paraplatin, cisplatin, ifosfamide, and paclitaxel, in patients in both early and advanced stage diseases [3]. While no survival benefit has been reported with the use of adjuvant radiotherapy, it decreases local recurrence [28]. However, in patients with stage I and II uterine carcinosarcoma who did not undergo lymph node dissection, there was a 21% reduction in death among those who underwent radiotherapy [29]. The primary treatment for ovarian carcinosarcoma is cytoreductive surgery, followed by platinum-based chemotherapy, usually carboplatin-paclitaxel [3]. Cases of carcinosarcoma in the paranasal sinus [10], salivary gland [30], esophagus [31], and urinary bladder [4] treated with surgical resection combined with radiotherapy or chemotherapy have also been reported, although there is still a lack of consensus on the mode of therapy owing to the limited number of cases. Orbital carcinosarcoma is a rare disease entity. Although it can be successfully treated, as in our case, a consensus on the treatment cannot be established due to the small number of cases. Future research on the genetic, molecular signaling pathways, and treatment responses of these tumors, incorporated from different body parts should be considered. Ideally, further well-designed nonrandomized studies should be performed with tumor resection following either chemotherapy, radiotherapy, or both.

## 4. Conclusions

Carcinosarcomas are rare but aggressive tumors that carry a risk of recurrence and often metastasize. When they occur in the orbits, they may invade the intracranial cavity, resulting in an increased mortality rate. We present a case of rare primary orbital carcinosarcoma that was successfully treated with surgical resection and radiotherapy. Therefore, the therapy of choice for such tumors occurring in different body tissues should be tumor resection for staging and possibly cytoreduction, followed by radiotherapy or chemotherapy. Physicians should be aware of the clinical presentation of this rare entity to ensure aggressive and successful medical management.

## Figures and Tables

**Figure 1 medicina-58-00317-f001:**
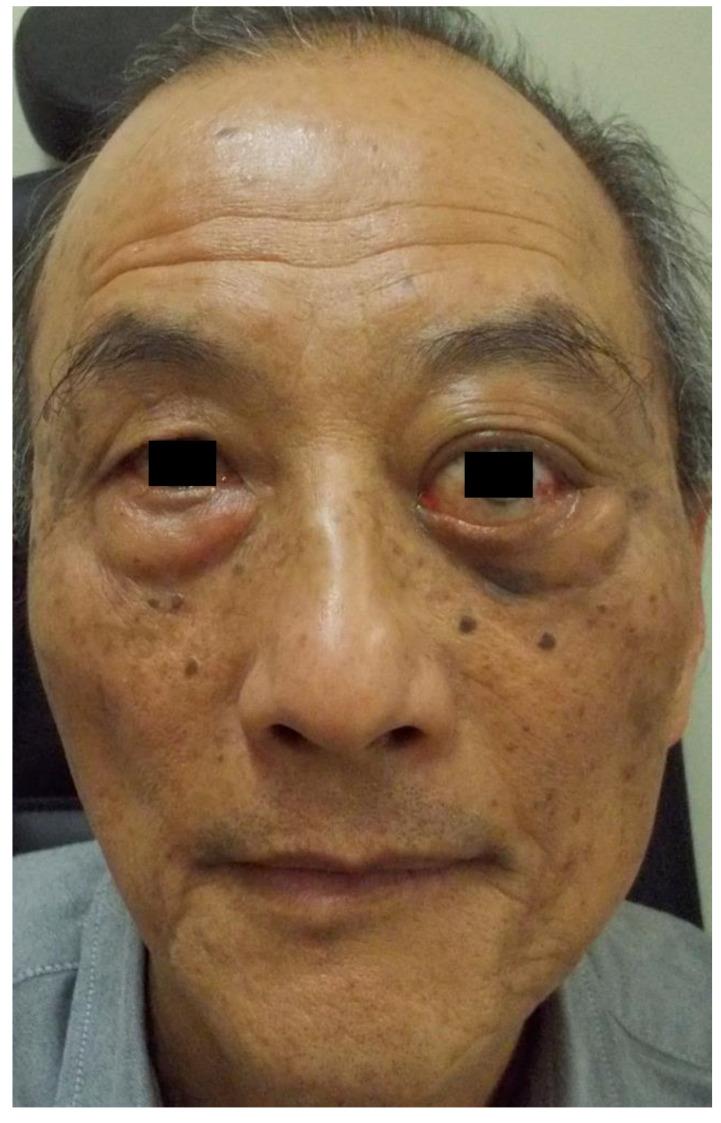
General appearance shows exophthalmos and lagophthalmos of the left eye.

**Figure 2 medicina-58-00317-f002:**
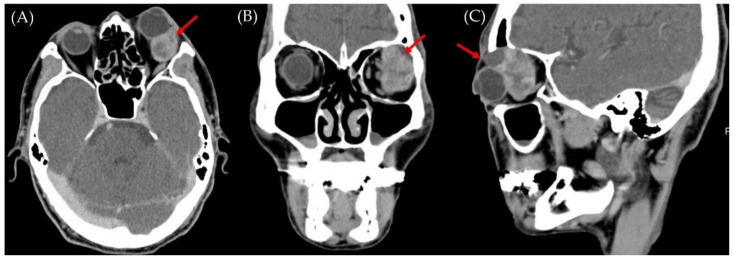
Orbital computed tomography scan reveals a heterogeneously enhancing mass (red arrows) in the left retrobulbar orbital cavity: (**A**) Axial plane; (**B**) coronal plane; (**C**) sagittal plane.

**Figure 3 medicina-58-00317-f003:**
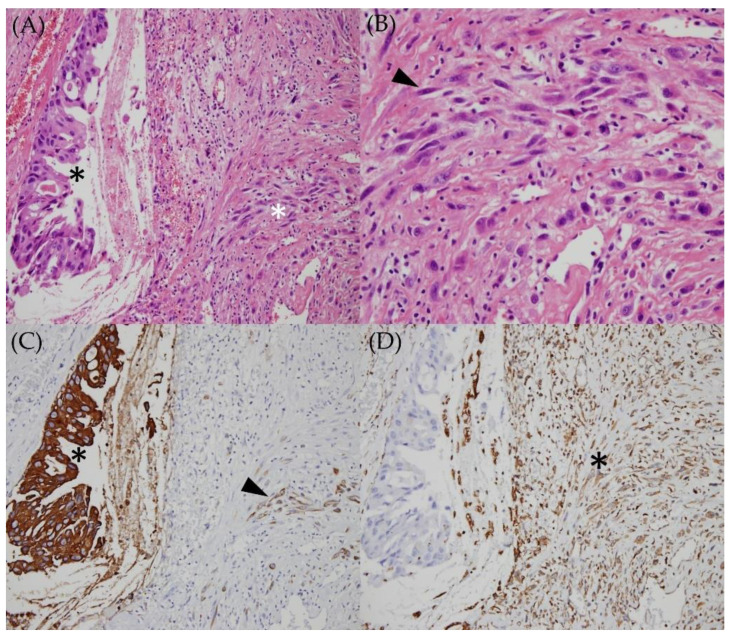
(**A**) The adenocarcinoma shows malignant glandular tumor cells with cribriform pattern (black asterisk), intermixed with sarcomatous spindle tumor cells (white asterisk). Hematoxylin-eosin stain, original magnification ×200; (**B**) sarcoma cells reveal pleomorphic, hyperchromatic, and spindle (arrowhead) to ovoid nuclei infiltrating the stroma. Hematoxylin-eosin stain, original magnification ×400. (**C**) glandular tumor cells (black asterisk) and focal sarcomatous cells (arrowhead) reveal positive pan-cytokeratin staining. Pan-cytokeratin stain, original magnification ×200 (**D**) immunohistochemical staining for vimentin shows a negative result for the glandular tumor cells and positive immunoactivity for the sarcomatous component (black asterisk). Vimentin stain, original magnification ×200.

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
