# Peer review of "Successful Resection of Retrobulbar Carcinosarcoma without Recurrence: A Case Report"

_medicina, 2022, doi:10.3390/medicina58020317_

Round 1
Reviewer 1 Report
The topic of this paper is interesting for large number of readers.
The findings from this paper is useful in everyday clinical practise. My opinion is that the paper shul be published
Reviewer 2 Report
Orbital carcinosarcoma is a rare and malignant mixed tumor. So far only five cases were reported regarding orbital carcinosarcoma. Owing to the rarity, there is no consensus on the management of this tumor. In this case report, the author presented an interesting orbital carcinosarcoma case with great prognosis comparing with previous reported cases. The medical record is quite explicit from the perspective of diagnosis and treatment. It is instructive for clinical practice. One minor point is that in Figure 1, please mask out (you can put a black band on) patients’ eyeballs for confidentiality.
Reviewer 3 Report
The aim of the article is to report a case of a very rare orbital tumor which presented with significant proptosis and limited eye movements. Its main contribution is to share the clinical experience regarding treatment and presentation since the incidence of presented case of orbital carcinosarcoma is extremely rare.
The case report is original and well defined. The results provide an advance in current knowledge since there are no relevant treatment guidelines available.
The English language is appropriate and understandable.
The manuscript is presented in a well structured manner.
References do not comprise self citations. The references are not within last 5 years but that is due to the specific rarity of the case presented.
The manuscript results could be reproducible. Images are appropriate and are easy to interpret.
- It is not clearly stated if the regular histopathologic specimen (not frozen section) surgical margins were free of tumor cells since EOM were compressed and tumor extension in the frontal sinus visible on CT imaging.... in that case different surgical approach
- PET was performed 1 month postoperatively and showed no systemic malignancy, but was there any additional workup for distant metastasis obtained in a follow up period (48 months)?
- MRI controls would not show local bone invasion.
Specific comments referring to line numbers:
Line 42.... Evidence of exophthalmos + hypotropia...
Line 45... Hypophasis – the term not commonly used in literature
Line 51... describing CT – anterior + downward displacement of the left eyeball
Line 172... probably is not possible to conduct prospective RT
